# Emergence of Thymidine-Dependent *Staphylococcus aureus* Small-Colony Variants in Cystic Fibrosis Patients in Southern Brazil

Dilair Camargo de Souza,[a] Laura Lucia Cogo,[a] Libera Maria Dalla-Costa,[b] Ana Paula de Oliveira Tomaz,[a,b] Daniele Conte,[b] Carlos Antonio Riedi,[c] Nelson Augusto Rosario Filho,[c] Jussara Kasuko Palmeiro[b,d]

aLaboratório de Bacteriologia, Complexo Hospital de Clínicas, Universidade Federal do Paraná, Curitiba, Paraná, Brazil
bFaculdades e Instituto de Pesquisa Pelé Pequeno Príncipe, Curitiba, Paraná, Brazil
cDepartamento de Pediatria, Complexo Hospital de Clínicas, Universidade Federal do Paraná, Curitiba, Paraná, Brazil
dDepartamento de Análises Clínicas, Centro de Ciências da Saúde, Universidade Federal de Santa Catarina, Florianópolis, Santa Catarina, Brazil

**ABSTRACT** We characterized *Staphylococcus aureus* small-colony variant (SCV) strains isolated from cystic fibrosis (CF) patients in southern Brazil. Smaller colonies of *S. aureus* were isolated from respiratory samples collected consecutively from 225 CF patients from July 2013 to November 2016. Two phenotypic methods—the auxotrophic classification and a modified method of antimicrobial susceptibility testing—were employed. PCR was conducted to detect the *mecA*, *ermA*, *ermB*, *ermC*, *msrA*, and *msrB* resistance genes. Furthermore, DNA sequencing was performed to determine the mutations in the *thyA* gene, and multilocus sequence typing was used to identify the genetic relatedness. *S. aureus* strains were isolated from 186 patients (82%); suggestive colonies of SCVs were obtained in 16 patients (8.6%). The clones CC1 (ST1, ST188, and ST2383), CC5 (ST5 and ST221), and ST398 were identified. Among SCVs, antimicrobial susceptibility testing showed that 77.7% of the isolates were resistant to multiple drugs, and all of them were susceptible to vancomycin. *mecA* (2), *ermA* (1), *ermB* (1), *ermC* (3), and *msrB* (18) were distributed among the isolates. Phenotypically thymidine-dependent isolates had different mutations in the *thyA* gene, and frameshift mutations were frequently observed. Of note, revertants showed nonconservative or conservative missense mutations. SCVs are rarely identified in routine laboratory tests.

**IMPORTANCE** Similar findings have not yet been reported in Brazil, emphasizing the importance of monitoring small-colony variants (SCVs). Altogether, our results highlight the need to improve detection methods and review antimicrobial therapy protocols in cystic fibrosis (CF) patients.

**KEYWORDS** SCVs, thymidine dependent, auxotrophism, *thyA*, MLST, antimicrobial susceptibility

*S*taphylococcus aureus is an important pathogen that can colonize and infect the airways of cystic fibrosis (CF) patients (1). Small-colony variants (SCVs) emerged in *S. aureus* as a phenotype related to chronic and recurrent infections (2) because of their ability to resist antimicrobial treatment and to persist in host cells; they show attenuated virulence and immune evasion (3, 4).

Identification of stable SCVs (sSCVs) is challenging in laboratory settings, because specific substrates, such as hemin, menadione, or thymidine, are required for their growth (5). Owing to this metabolic deficiency, sSCVs are 10-fold smaller than normal colonies, nonpigmented, and nonhemolytic and can adhere to the agar surface (3). They may also display false-negative results for catalase and coagulase tests (2) and fail

**Citation** de Souza DC, Cogo LL, Dalla-Costa LM, Tomaz APDO, Conte D, Riedi CA, Rosario Filho NA, Palmeiro JK. 2021. Emergence of thymidine-dependent *Staphylococcus aureus* small-colony variants in cystic fibrosis patients in Southern Brazil. Microbiol Spectr 9:e00614-21. https://doi.org/10.1128/Spectrum.00614-21.

Address correspondence to Jussara Kasuko Palmeiro, jukasuko@gmail.com.

to grow on Mueller-Hinton agar (MHA) (3). In addition to sSCVs, other phenotypic changes in *S. aureus* include revertant types (unstable SCVs), which are also difficult to identify, especially when they do not grow in the form of SCVs. The reversal mechanism is unclear; however, it has been proposed as the key to successful reinfection of the host (4, 5). Thus, persistent *S. aureus* infections may be caused by a variety of phenotypes, even within a genetically clonal population (6).

Thymidine auxotrophism has emerged as a consequence of genetic mutations in the *thyA* gene due to prolonged treatment with trimethoprim-sulfamethoxazole (7, 8). This antibiotic inhibits the synthesis of tetrahydrofolic acid, which serves as a cofactor for thymidylate synthase (encoded by *thyA*), an enzyme required for the conversion of uracil into thymidine (3). Owing to these mutations, thymidine-dependent SCVs (TD-SCVs) show intrinsic resistance to trimethoprim-sulfamethoxazole (TMP-SMX) (7, 9). *S. aureus* TD-SCVs have been associated with different chronic infections such as endocarditis, arthritis, osteomyelitis, and rhinosinusitis (3). Remarkably, despite their low virulence, they were previously isolated in the context of bacteremia (8).

The clinical implications of SCVs and their difficult microbiological diagnosis highlight the need for the improvement of the understanding of their growth characteristics, as well as the development of new culture identification and antimicrobial susceptibility testing methods. Of note, such limitations may lead to misdiagnoses and, consequently, therapeutic failure. In this study, we describe the prevalence, as well as the phenotypic and molecular characteristics of *S. aureus* SCVs isolated from CF patients in the community. Notably, *S. aureus* TD-SCVs have never been reported in Brazil, emphasizing the need for improved monitoring in the context of this pathogen.

## RESULTS

**Isolation of *S. aureus* SCVs: bacterial identification and prevalence.** A total of 225 respiratory samples from the same number of CF patients were referred for microbiological analyses. The numbers of male and female patients were similar ($n = 117$, 52%, and $n = 108$, 48%, respectively). The median age of the patients was 6 years (range, 1 month to 70 years). We isolated *S. aureus* from 186 patients (82%) and found suggestive colonies of SCVs in 16 patients (8.6%). The median ages of patients with normal and SCV *S. aureus* isolates were 5 (range, 1 to 70 years) and 13 (range, 5 to 57 years) years, respectively.

One bacterial isolate was studied from all patients except two patients with two isolates each (patient 7 [P7], samples 7 and 17 [Sa7 and Sa17]; and P8, Sa8 and Sa18) (Table 1); therefore, 18 SCV isolates of *S. aureus* were characterized. Table 1 summarizes the bacterial identification, the antimicrobial susceptibility profile, and the detection of resistance genes. Overall, the standard biochemical characteristics indicated *S. aureus*, except for 11 isolates that did not grow on 7% NaCl agar and MHA. Importantly, Vitek mass spectrometry (MS) and the amplification of the *nuc* gene, encoding a specific thermonuclease of *S. aureus*, identified all suspected SCV colonies as *S. aureus*; however, Vitek 2 did not show the same accuracy. Five isolates were identified as coagulase-negative staphylococci species instead of *S. aureus*.

**Antimicrobial susceptibility and $\beta$-lactam/MLS$_B$ resistance genes.** Most isolates (except for Sa3, Sa5, Sa11, and Sa14) presented MICs with reduced susceptibility for at least three classes of antimicrobials (77.7%, $n = 14$). Only a unique isolate was sensitive to trimethoprim-sulfamethoxazole (Sa12) and to clindamycin and erythromycin (Sa14). Ciprofloxacin showed activity against SCV isolates, of which 61% ($n = 11$) were susceptible and 31.3% ($n = 5$) were intermediate to this antimicrobial. Only oxacillin and vancomycin showed good activity against these isolates (Table 1).

In oxacillin-resistant isolates, we detected the *mecA* gene. Additionally, for isolates showing macrolide-lincosamide-streptogramin B (MLS$_B$) resistance, constitutive and inducible MLS$_B$ phenotypes ($n = 11$ and $n = 6$, respectively) showing different genotypes were identified, including the coexpression of *erm* and *msr* or the single detection of *msrB*. Three isolates displayed positive results for the inducible MLS$_B$ test (D-test), although the *erm* gene was not detected (Sa5, Sa11, and Sa18). Of note, one isolate positive for the *msrB*

**TABLE 1** Biochemical and antimicrobial resistance features of SCV isolates

| Patient | Isolate ID[a] | Bacterial identification methods[b] | | | | | | MIC (mg/liter)[c] | | | | | | Inducible MLS$_B$[d] test (D-test) | Resistance genes | | |
|---|---|---|---|---|---|---|---|---|---|---|---|---|---|---|---|---|---|
| | | Biochemical standard | | | Automated systems | | nuc gene | | | | | | | | β-Lactams (mecA) | MLS$_B$ | |
| | | Catalase | 7% NaCl | Coagulase | Vitek 2 | Vitek MS | | CIP | CLI | ERY | TMP/SMX | VAN | OXA | | | | msrB |
| P1 | Sa1 | + | – | + | S. aureus | S. aureus | + | 2 | 0.06 | >32 | >16/304 | 1 | >16 | + | + | ermC | msrB |
| P2 | Sa2 | + | – | + | S. aureus | S. aureus | + | 8 | >16 | >32 | >16/304 | 0.5 | 0.25 | – | – | ermB | msrB |
| P3 | Sa3 | + | – | + | S. aureus | S. aureus | + | 2 | 0.06 | 32 | >16/304 | 2 | 0.25 | + | – | ermA | msrB |
| P4 | Sa4 | + | – | + | S. capitis/warneri | S. aureus | + | 4 | >16 | >32 | >16/304 | 0.25 | 0.125 | – | – | | msrB |
| P5 | Sa5 | + | – | + | S. lentus | S. aureus | + | 1 | 0.125 | >32 | 16/304 | 2 | 0.5 | + | – | | msrB |
| P6 | Sa6 | + | – | + | S. aureus | S. aureus | + | 4 | >16 | >32 | >16/304 | 1 | 1 | – | – | | msrB |
| P7 | Sa7 | + | – | + | S. lentus | S. aureus | + | 1 | 4 | 16 | >16/304 | 1 | 0.5 | – | – | | msrB |
| P8 | Sa8 | + | – | + | S. aureus | S. aureus | + | 2 | >16 | >32 | >16/304 | 2 | 1 | – | – | | msrB |
| P9 | Sa9 | + | – | + | S. lentus | S. aureus | + | 8 | >16 | >32 | >16/304 | 1 | 2 | + | – | ermC | msrB |
| P10 | Sa10 | + | – | + | S. aureus | S. aureus | + | 4 | >16 | >32 | >16/304 | 1 | 1 | – | – | | msrB |
| P11 | Sa11 | + | – | + | S. aureus | S. aureus | + | 1 | 0.06 | >32 | >16/304 | 1 | 0.125 | + | – | | msrB |
| P12 | Sa12 | + | w | + | S. warneri | S. aureus | + | 4 | 0.5 | >32 | 2/38 | 1 | >16 | + | + | ermC | msrB |
| P13 | Sa13 | + | w | + | S. aureus | S. aureus | + | 1 | >16 | >32 | >16/304 | 1 | 2 | – | – | | msrB |
| P14 | Sa14 | + | w | + | S. aureus | S. aureus | + | 1 | 0.25 | 0.25 | 8/152 | 1 | 1 | – | – | | msrB |
| P15 | Sa15 | + | w | + | S. aureus | S. aureus | + | 1 | >16 | >32 | 4/76 | 2 | 2 | – | – | | msrB |
| P16 | Sa16 | + | w | + | S. aureus | S. aureus | + | 4 | 0.25 | >32 | >16/304 | 1 | 0.5 | + | – | | msrB |
| P7 | Sa17 | + | w | + | S. aureus | S. aureus | + | 2 | 8 | >32 | >16/304 | 2 | 0.5 | – | – | | msrB |
| P8 | Sa18 | + | w | + | S. aureus | S. aureus | + | 2 | >16 | >32 | 4/76 | 2 | 1 | – | – | | msrB |
| QC[e] | | | | | | | | 0.25 | 0.25 | 0.25 | 0.5/9.5 | 0.5 | 0.25 | | | | |
| QC[f] | | | | | | | | | | | 0.5/9.5 | | | | | | |

[a]ID, identifier; Sa7 and Sa17 were isolated from the same patient; Sa8 and Sa18 were also isolated from the same patient (a different one).
[b]w, weak growth.
[c]CIP, ciprofloxacin; CLI, clindamycin; ERY, erythromycin; TMP/SMX, trimethoprim-sulfamethoxazole; VAN, vancomycin; OXA, oxacillin.
[d]MLS$_B$, macrolide-lincosamide-streptogramin B resistance.
[e]QC, quality control results. S. aureus ATCC 29213.
[f]E. faecalis ATCC 29212.

gene still showed susceptibility to MLS$_B$ (Sa14). Moreover, one isolate showed high MICs to clindamycin and erythromycin in addition to the positive D-test (Sa9) (Table 1).

**Phenotypic screening and molecular characterization of the auxotrophism of *S. aureus* SCVs.** The 18 suggestive colonies of SCVs were subjected to nutritional dependence assays. Eleven isolates from different patients grew only on MHA when supplemented with thymidine (in both tests, MHA and disk supplemented). The remaining seven isolates grew poorly on nonsupplemented MHA and MHA supplemented with thymidine, menadione, and hemin and were, therefore, characterized as revertant SCVs (Table 2). No isolate showed dependency on menadione or hemin. Figure 1 shows examples of the morphology of normal colonies and SCVs grown on mannitol salt agar (MSA) and blood agar, their Gram morphologies, and their auxotrophism characterization.

Regarding the sequencing of the *thyA* gene, among the isolates phenotypically determined as SCVs, most showed insertion-deletion (indel) mutations, while revertant isolates showed nonsynonymous and synonymous mutations. We assigned genotypes according to the types of mutations identified in the *thyA* gene: (i) sSCVs included isolates that showed frameshift mutations caused by indels (7 isolates) or nonsense mutations resulting in a premature stop codon (1 isolate); (ii) revertants comprised missense conservative (1 isolate) and nonconservative mutations around the dUMP-binding site (2 isolates); and (iii) the wild type (WT) contained silent point mutations (7 isolates) or no mutations at all (1 isolate) (Table 2). Two isolates phenotypically determined as sSCVs (Sa2 and Sa8) and almost all revertant isolates showed the WT genotype; the only revertant isolate that did not was classified as sSCV (Sa13). Concerning the two patients with two isolates each (P7 [Sa7 and Sa17] and P8 [Sa8 and Sa18]), when comparing the isolates, we observed different mutations in only one (Table 2).

**Diversity of *S. aureus* phenotypes over time, treatment schedules, and molecular typing.** Figure 2 shows the diversity of *S. aureus* phenotypes isolated during the study in the 16 patients from whom SCVs were identified through phenotypic testing, starting from 1 year before to 1 year after the emergence of the first SCV; the use of TMP-SMX and azithromycin (AZT) during this period is also represented. Overall, monotherapy or combined therapy (AZT and TMP-SXT) was used for a long time in all patients. Of note, in four patients (P2, P3, P4, and P6), data on treatment were not available, and one patient (P14) did not receive antimicrobial therapy because he was asymptomatic. Most patients showed at least two different colony variants during the evaluation period, highlighting the heterogeneity in patients subjected to combination therapy.

The multilocus sequence typing (MLST) results are presented in Table 2. Molecular typing was performed only in the genotypically isolates classified as sSCVs. Clonal complex 5 (CC5) was found in three patients, with two isolates belonging to sequence type 5 (ST5). CC1 was also detected in three patients but with different sequence types, and ST398 was detected in two patients; in another two patients, the CC was nontypeable.

## DISCUSSION

The SCV phenotype in the context of *S. aureus* has attracted much attention in the last 2 decades because it is related to chronic infections and difficult treatments (10). The present study is the first to characterize SCVs isolated from the respiratory tracts of CF patients in Brazil.

Due to the metabolic changes observed in *S. aureus* that determine the emergence of SCV colonies, the identification of this phenotype is challenging. In our study, the standard biochemical analysis showed the expected results. However, biochemical changes can occur and may lead to delayed or negative test results (3). Remarkably, the use of 7% NaCl medium contributed to the initial identification; wild-type *S. aureus* tended to grow usually, and SCVs did not grow or grew weakly compared to the phenotype SCV that usually grows in MSA medium (11). Of note, the use of 7% NaCl agar, used in this study, was not reported before. We used a modified Chapman medium (MSA) that does not contain mannitol and phenol red as a pH indicator.

When the morphology suggests *S. aureus* SCVs but the classical tests do not confirm it, other methodologies such as screening of the *nuc* gene should be used (12). The Vitek 2 Compact automated system only provided correct identification of 72.2% of

**TABLE 2** Screening for nutritional dependence, molecular typing, and *thyA* mutations in thymidine-dependent and revertant *Staphylococcus aureus* SCV isolates

| Patient | Isolate ID[a] | Agar/disk supplementation[b] | | | | Phenotype | thyA sequencing | | | Genotype[c] | Multilocus sequence typing[d] | |
|---|---|---|---|---|---|---|---|---|---|---|---|---|
| | | WS | THY | HEM | MEN | | Synonymous mutation | Nonsynonymous mutation | Alteration(s) | | Clonal complex | Sequence type |
| P1 | Sa1 | −/− | +/+ | −/− | −/− | sSCV | A45G | A617G | Missense nonconservative mutation (Gln206Arg) | Revertant | 5 | 5 |
| P2 | Sa2 | −/− | +/+ | −/− | −/− | sSCV | A183G, C219T, A237G, T411A, A438T, T450G, T501A, A516G, T582C, C609A, C837T, C864T, T876A, A924C | | Silent point mutations | WT | NP | NP |
| P3 | Sa3 | −/− | +/+ | −/− | −/− | sSCV | T47G | Δ48TTTAGAAATAGG59 | Missense conservative mutation (Val16Gly)<br>Frameshift mutation (−12 nt[e] from amino acid 17) | sSCV | 1 | 2383 |
| P4 | Sa4 | −/− | +/+ | −/− | −/− | sSCV | A45G | CAACTT115CAACTCTT | Silent point mutation<br>Frameshift mutation (+2 nt from amino acid 40) | sSCV | 5 | 221 |
| P5 | Sa5 | −/− | +/+ | −/− | −/− | sSCV | A60G, C219T, A237G, T411A, A456G, A516G, T582C | A55C | Missense conservative mutation (Ile19Leu)<br>Silent point mutations | sSCV | 398 | 398 |
| P6 | Sa6 | −/− | +/+ | −/− | −/− | sSCV | | Δ590CACTTCCGCCTT601<br>Δ589GCACTTCCGCCTT601 | Frameshift mutation (−12 nt from amino acid 197)<br>Frameshift mutation (−13 nt from amino acid 197) | sSCV | 5 | 5 |
| P7 | Sa7 | −/− | +/+ | −/− | −/− | sSCV | G99A | A650G | Missense nonconservative mutation (Gln217Arg) | Revertant | 1 | 188 |
| P8 | Sa8 | −/− | +/+ | −/− | −/− | sSCV | T489C, A510G, A516G | | Silent point mutations | WT | NP | NP |
| P9 | Sa9 | −/− | +/+ | −/− | −/− | sSCV | A60G, A147T, C219T, A237G, T411A, A456G, A516G, T582C, A591T, A744C | A55C | Missense conservative mutation (Ile19Leu)<br>Silent point mutations | sSCV | 398 | 398 |
| P10 | Sa10 | −/− | +/+ | −/− | −/− | sSCV | | Δ766ATACATTTGGAG778<br>Δ591ACTTCCGCCTT601 | Frameshift mutation (−12 nt from amino acid 256)<br>Frameshift mutation (−11 nt from amino acid 198) | sSCV | NT | NT |
| P11 | Sa11 | −/− | +/+ | −/− | −/− | sSCV | | A192T | Missense non−conservative mutation (Leu64Phe) | sSCV | 1 | 1 |

**TABLE 2** (Continued)

| Patient | Isolate ID[a] | Agar/disk supplementation[b] | | | | Phenotype | thyA sequencing | | Alteration(s) | Genotype[c] | Multilocus sequence typing[d] | |
| | | WS | THY | HEM | MEN | | Synonymous mutation | Nonsynonymous mutation | | | Clonal complex | Sequence type |
|---|---|---|---|---|---|---|---|---|---|---|---|---|
| P12 | Sa12 | w/w | w/w | w/w | w/w | Revertant | A183G, C219T, A237G, T411A, A456G, A516G, C609A, C627T, A720G, C870T, C882T | Δ197G198 | Frameshift mutation (−1 nt from amino acid 66); Silent point mutations | WT | NP | NP |
| P13 | Sa13 | w/w | w/w | w/w | w/w | Revertant | | AAA451TAA | Nonsense mutation (Lys150Stop) | sSCV | NT | NT |
| P14 | Sa14 | w/w | w/w | w/w | w/w | Revertant | A183G, C219T, A237G, T411A, A456G, A516G, C609A, C627T, A720G, C870T, C882T | | Silent point mutations | WT | NP | NP |
| P15 | Sa15 | w/w | w/w | w/w | w/w | Revertant | | | No mutation | WT | NP | NP |
| P16 | Sa16 | w/w | w/w | w/w | w/w | Revertant | G99A | | Silent point mutations | WT | NP | NP |
| P7 | Sa17 | w/w | w/w | w/w | w/w | Revertant | G99A, A282C, T495A | T495A | Missense conservative mutation (Asp165Glu) | Revertant | NP | NP |
| P8 | Sa18 | w/w | w/w | w/w | w/w | Revertant | T489C, A510G, A516G | | Silent point mutations | WT | NP | NP |

[a]ID, Identifier; Sa7 and Sa17 were isolated from the same patient; Sa8 and Sa18 were also isolated from the same patient (a different one).
[b]WS, without supplementation; THY, thymidine; HEM, hemin; MEN, menadione; w, weak growth.
[c]WT, wild type.
[d]NT, nontypeable; NP, analysis not performed.
[e]nt, nucleotide(s).

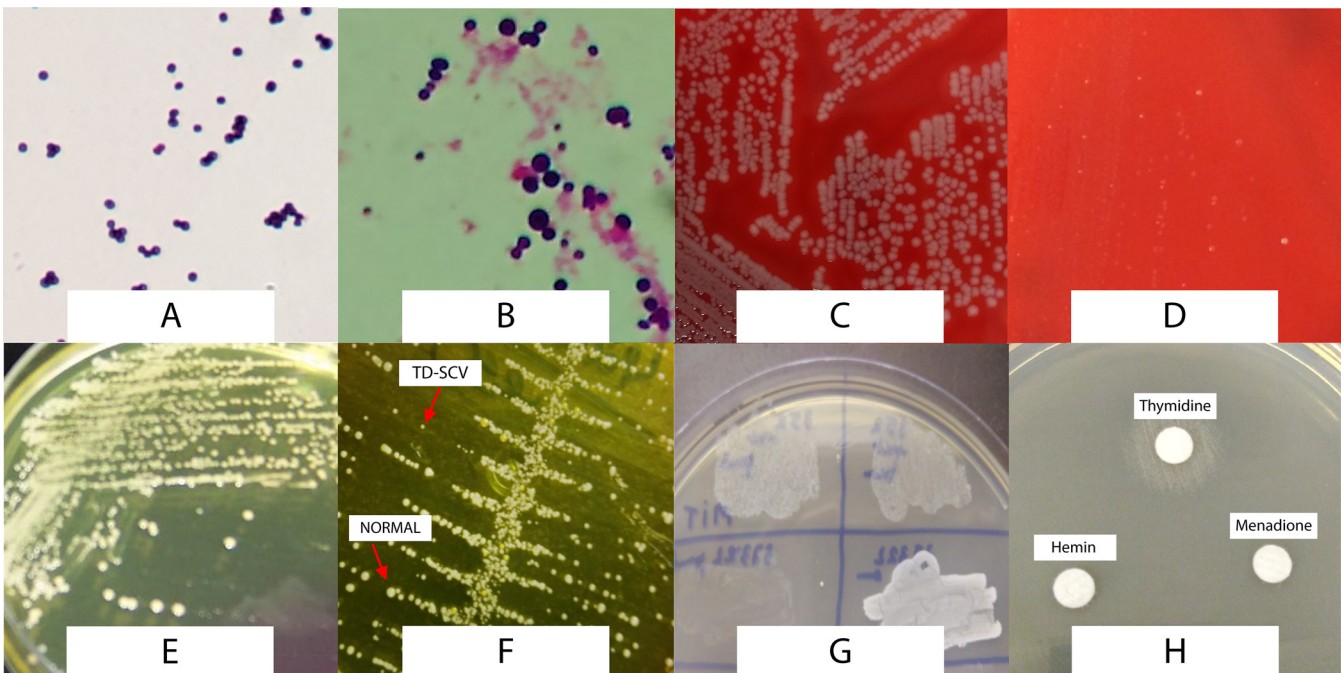

**FIG 1** Microbiological characteristics of clinical *Staphylococcus aureus* isolates. (A) Gram morphology of normal *S. aureus*. (B) Gram morphology of TD-SCV exhibiting an increase in cell size. (C) *S. aureus* isolates grown on Columbia blood agar. (D) TD-SCV grown on Columbia blood agar displaying smaller and nonhemolytic colonies. (E) Normal *S. aureus* isolates grown on mannitol salt agar. (F) Normal and TD-SCV isolates grown on mannitol salt agar showing different colony sizes. (G) Types of growth of TD-SCVs in the surface of Mueller-Hinton agar supplemented with thymidine (100 μg/ml). (H) Growth of TD-SCVs around the disc impregnated with thymidine placed onto Mueller-Hinton agar.

the *S. aureus* SCV isolates. Indeed, these data demonstrate that, due to decreased metabolism of SCVs, the use of automated colorimetric systems for their identification is limited, even for essential tests such as catalase and coagulase tests (2). On the other hand, the use of mass spectrometry (matrix-assisted laser desorption ionization–time of flight [MALDI-TOF]) would have been a better alternative to confirm the phenotypic identification in our study, as also described by Ota et al. (13).

Worrisome enough, most of our isolates showed resistance to at least three classes of antimicrobials, being classified as multidrug resistant (14). Expectedly, almost all SCV isolates were resistant to TMP-SMX (a feature of this phenotype) (9, 15). In fact, only one isolate was susceptible to this antimicrobial; of note, it was classified as a revertant, which can explain this finding. Although TMP-SMX is an antimicrobial of choice for the treatment of *S. aureus* infections in CF patients, continuous exposure to this drug is associated with the appearance of TD-SCVs (7). The drug impacts the bacterial folate pathway, inhibiting the production of two proteins involved in the synthesis of tetrahydrofolic acid, which acts as a cofactor for thymidylate synthase, resulting in mutations in the *thyA* gene and consequent resistance (3). Importantly, all patients with TD-SCVs were administered TMP-SMX; however, a few did not receive this antimicrobial for more than 1 year before the emergence of this phenotype.

We also observed elevated resistance to erythromycin and clindamycin in this study. Our results, however, are not in line with those of other studies reporting intermediate or low resistance (16, 17). We need to keep in mind that the macrolide azithromycin has anti-inflammatory properties and is used in CF patients to improve their lung function (18). The use of this antibiotic is associated with resistance to erythromycin and clindamycin, supporting our results (19). The gene *msrB*, found in all isolates, is related to the expression of efflux pumps, an additional mechanism associated with resistance to macrolides and streptogramin B; however, this mechanism is not associated with induced or constitutive resistance to clindamycin (20). Interestingly, the only isolate susceptible to macrolides (Table 1) was detected in a patient who did not use azithromycin (Fig. 2). Additionally, Table 1 shows the results of both constitutive and

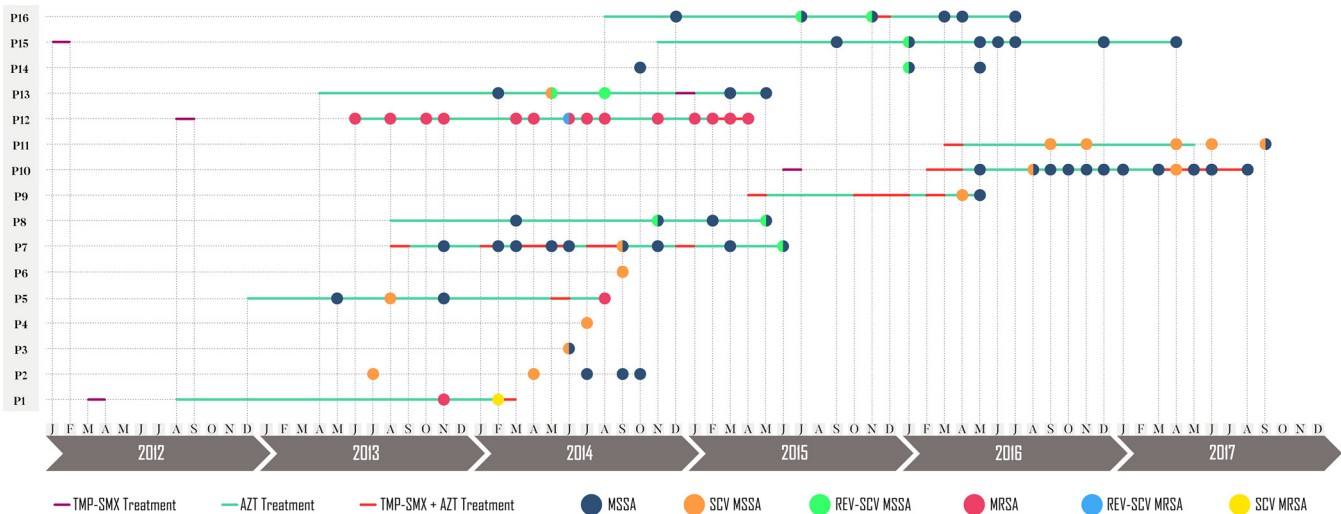

**FIG 2** Timeline of the isolation of different *Staphylococcus aureus* phenotypes, according to the use of antimicrobials and molecular typing. TMP/SMX, trimethoprim-sulfamethoxazole treatment; AZT, azithromycin treatment; TMP/SMX+AZT, trimethoprim-sulfamethoxazole plus azithromycin treatment.

inducible resistance (cMLS$_B$ and iMLS$_B$, respectively). Some isolates showing a positive phenotypic test (D-test) result did not reveal a matching genotype (the amplification of *erm* genes). Of note, here, only the most frequently found *ermA*, *ermB*, and *ermC* genes were screened for; the presence of less frequently described genes, such as *ermF*, *ermY* (21), and *ermT* (22), was not investigated.

A minority of isolates were oxacillin resistant, as confirmed by the expression of the *mecA* gene, as previously reported (16, 23). Of note, methicillin-resistant *S. aureus* is associated with worsening lung functions and the subsequent mortality risk in CF patients (24). Therefore, the emergence of methicillin-resistant *S. aureus* (MRSA) SCV isolates can further aggravate the condition of infected patients (25). The susceptibility to vancomycin observed here may be related to the restricted use of this antimicrobial in the hospital environment.

Regarding the nutritional dependence, all isolates were TD-SCVs. This auxotrophism is most frequent in respiratory samples from CF patients (7). In line with our results, different studies showing only thymidine dependence have been reported (4, 26); however, codependence on hemin and menadione has also been described (16, 27). We found a prevalence of 8.6% of SCVs among the patients included, which is within the ranges reported in different countries. For instance, the prevalence of SCVs in different European countries ranges from 4% to 33% (3), in the United States, from 4.6% to 24% (27, 28), and in Turkey, from 8% to 21% (17, 23).

Sequencing of the *thyA* gene in TD-SCVs has revealed several mutations. Here, we show that frameshift mutations (mostly deletions) resulted in the emergence of the sSCV phenotype, while missense mutations were associated with the revertant phenotype. Mutations can occur anywhere in the gene, in the beginning, end, or near or at the active site of the enzyme (4, 15, 29). Owing to the complex characteristics of TD-SCVs (particularly of the unstable phenotype, which is not yet fully understood), further studies are still needed to understand better whether additional mutations can act in a compensatory or stabilizing manner to restore the activity of the enzyme thymidylate synthase, leading to revertant and normal phenotypes.

The isolates Sa3, Sa4, Sa5, Sa6, Sa9, Sa10, and Sa11 were phenotypically and genetically determined as sSCVs; in subcultures after initial isolation, the phenotype remained stable. Although these isolates may be less fit because of the mutations behind this phenotype, they can survive owing to the presence of thymidine in secretions in the airways of CF patients (4, 11). Of note, sSCV isolates that presented missense nonconservative mutations (isolates Sa1 and Sa7) and the revertant isolate with missense conservative mutations (Sa17) showed, after subculturing, reversible and

**TABLE 3** List of primers and PCR conditions used in this study

| Gene | Sequence (5′→3′)[a] | Fragment size (bp) | PCR conditions | Reference |
|------|---------------------|--------------------|----------------|-----------|
| *Nuc* | F GCCACGTCCATATTTATCAG<br>R TATGGTCCTGAAGCAAGTG | 117 | 94°C for 1 min, 52°C for 30 s, 72°C for 30 s | 36 |
| *thyA* | F GCAATGACTACACTGCTATTGG<br>R GAGGTGTTATCGCATATGTTG | 957 | 94°C for 45 s, 56°C for 45 s, 72°C for 45 s | This study |
| *mecA* | F TCCAGATTACAACTTCACCAGG<br>R CCACTTCATATCTTGTAACG | 162 | 94°C for 1 min, 52°C for 30 s, 72°C for 30 s | 37 |
| *ermA* | F TCTAAAAAGCATGTAAAAGAA<br>R CTTCGATAGTTTATTAATATTAGT | 645 | 94°C for 30 s, 52°C for 1 min, 72°C for 90 s | 38 |
| *ermB* | F GAGTGAAAAGGTACTCAACCAAATAA<br>R TTGGTGAATTAAAGTGACACGAA | 208 | 94°C for 30 s, 52°C for 1 min, 72°C for 90 s | 39 |
| *ermC* | F TCAAAACATAATATAGATAAA<br>R GCTAATATTGTTTAAATCGTCAAT | 642 | 94°C for 30 s, 47°C for 1 min, 72°C for 90 s | 40 |
| *msrA* | F TATAGCGCTCGTAGGTGCAA<br>R GTTCTTTCCCCACCACTCAA | 270 | 94°C for 1 min, 52°C for 30 s, 72°C for 30 s | 39 |
| *msrB* | F TGTGGATGGCCTAGCTTTTC<br>R TCGCCATAACCCAATTCTTC | 230 | 94°C for 1 min, 52°C for 30 s, 72°C for 30 s | 39 |

[a]F, forward; R, reverse.

normal phenotypes, respectively. In line previous reports (4), these results suggest that unstable phenotypes may be lost. Curiously, it was not possible to characterize SCVs via phenotypic tests alone, as exemplified by the Sa13 isolate. The predominance of reversible colonies made it impossible to detect the sSCV phenotype. This fact highlights the need for the use of highly accurate assays, such as sequencing.

Remarkably, in our study, the emergence of TD-SCV was often related to some type of mutation in the *thyA* gene. However, in isolates Sa2 and Sa8, with the stable phenotype, and in isolates Sa12, Sa14, Sa16, and Sa18, with the revertant phenotype, only silent point mutations and the wild genotype were observed. Of note, in Sa15, no mutations (synonymous or nonsynonymous) were observed. In these isolates, the reversible SCV phenotype may be due to other mutations in the genome of *S. aureus* and not in the *thyA* region (4). This is important for understanding the mechanisms behind the appearance of SCVs and the pathogenesis of *S. aureus* (30). Notably, the unstable phenotype seems to play a major role in the relapse of seemingly defeated infections (27). Multi-omics approaches must, therefore, be used to increase the understanding of these strains.

Finally, clonal analysis was performed only for sSCVs to verify if there was a common clone; however, genetic variability was detected among the isolates. The ST5 and ST398 clones were found more than once. These clones have been previously detected in the context of methicillin-susceptible *S. aureus* (MSSA) and MRSA isolates with the SCV phenotype from respiratory samples of CF patients (31); the ST5 clone was described as a community-acquired isolate (32), and ST398 has been reported in animal and human infections (8) and may present high virulence (33).

Altogether, our data highlight the challenges in the recognition and identification of *S. aureus* SCVs. Lack of accurate methods in laboratory settings interferes directly in the recognition and identification of these variants. Therefore, their local (and even global) prevalence, we believe, is expressive and underestimated. Additionally, our results emphasize the potential treatment challenges in the context of these particular strains, since all of the isolates were multidrug resistant.

## MATERIALS AND METHODS

**Study settings and ethics statement.** This study was performed at Complexo Hospital de Clínicas, Universidade Federal do Paraná (CHC/UFPR), an academic care hospital located in Curitiba, Paraná, southern Brazil. The Institutional Ethics Review Board of CHC/UFPR approved this study under reference number CAAE 45.063115.90000.0096.

**Clinical sample collection, bacterial identification, and phenotypic characterization of SCVs.** A total of 225 respiratory samples (sputum, bronchoalveolar lavage fluid, and oropharyngeal swab samples) from CF patients at CHC/UFPR were collected and analyzed consecutively during July 2013 and November 2016. Mannitol salt agar (MSA) was used to isolate *S. aureus* under aerobic conditions. Smaller colonies were identified using standard biochemical tests. All isolates were stored at −80°C in

Trypticase soy broth (TSB; HiMedia, Mumbai, India) containing 15% glycerol. Bacterial identification was performed using 7% NaCl agar, the Vitek 2 Compact instrument, and Vitek MS (bioMérieux S.A., Marcy l'Etoile, France), as per the manufacturer's instructions. Amplification of the *nuc* gene via PCR was performed for confirmation; the primers and amplification conditions used are indicated in Table 3.

The nutritional dependence of *S. aureus* SCVs was further assessed. Isolates (0.5 McFarland standard) were inoculated on (i) Mueller-Hinton agar (MHA) (Oxoid, Thermo Fisher Scientific, Waltham, MA, USA) supplemented with hemin (10 $\mu$g/ml), menadione (25 $\mu$g/ml), or thymidine (100 $\mu$g/ml) (Sigma-Aldrich, Merck, St. Louis, MO, USA), and (ii) MHA containing blank discs impregnated with 15 $\mu$l of each of the above-mentioned solutions (34), both incubated under aerobic conditions at 35°C for 24 to 72 h. *S. aureus* SCV isolates were characterized as nutritionally dependent when there was growth on a particular substrate but there was no growth in its absence. Additionally, isolates were considered revertants when they showed small colonies and grew weakly in the presence and absence of substrates on MHA.

**Antimicrobial susceptibility testing and detection of $\beta$-lactam, macrolide, lincosamide, and streptogramin B resistance genes.** TD-SCV and revertant isolates, as defined by the phenotypic methods mentioned above, were tested for susceptibility to ciprofloxacin, clindamycin, erythromycin, oxacillin, trimethoprim-sulfamethoxazole, and vancomycin by broth dilution using brain heart infusion broth (BHI; Oxoid, Thermo Fisher Scientific) supplemented with thymidine (100 $\mu$g/ml) (23, 34). *S. aureus* ATCC 29213 was used as a control strain; test was performed using cation-adjusted Muller-Hinton broth (CAMHB; Oxoid, Thermo Fisher Scientific). To evaluate the interference of thymidine (used to supplement the medium for susceptibility testing and the possible effect on TMP-SXT results) on bacterial growth, quality control was also performed with *Enterococcus faecalis* ATCC 29212 (Table 1). The double-disk diffusion method with clindamycin and erythromycin disks was performed to determine MLS$_B$ resistance phenotypes, using MHA supplemented with thymidine (100 $\mu$g/ml). Test results were interpreted according to the CLSI standards (CLSI M100S-ED26:2016; https://clsi.org).

The *mecA* gene (encoding oxacillin [OXA] resistance determinant) and the *ermA, ermB, ermC, msrA,* and *msrB* genes (encoding MLS$_B$ resistance determinants) were detected via PCR using specific primers and conditions previously described (Table 3).

***thyA* sequencing and molecular typing.** Phenotypically TD-SCV stable and revertant *S. aureus* isolates were tested for the presence of mutations in *thyA* by using PCR and sequencing (Table 3). The PCR products were sequenced using a 3730XL DNA analyzer (Applied Biosystems, Carlsbad, CA, USA). Nucleotide and protein sequences were compared to the sequences of eight *S. aureus* strains, available in the GenBank database: (i) COL (accession no. NC_002951.2), (ii) MRSA252 (NC_002952.2), (iii) MSSA476 (NC_002953.3), (iv) Mu50 (NC_002758.2), (v) MW2 (NC_003923.1), (vi) N315 (NC_002745.2), (vii) NCTC8375 (NC_007795.1), and (viii) USA300_FPR3757 (NC_007793.1). We also included the *thyA* sequencing results of our wild-type isolates in the analysis.

Multilocus sequence typing (MLST) was performed by PCR amplification and sequencing of seven *S. aureus* housekeeping genes (*arcC, aroE, glpF, gmk, pta, tpi,* and *yqiL*) only for genetically confirmed SCVs, using previously described primers and procedures (35). Sequences were analyzed using the *S. aureus* MLST website (https://pubmlst.org/saureus/).

## ACKNOWLEDGMENTS

We thank the Central Laboratory of Paraná (LACEN), Paraná, Brazil, for the conduction of the MALDI-TOF assay. We also thank the staff of the Life Sciences Core Facility (GOSeq Biotecnologia), Federal University of Paraná (UFPR), for help with DNA sequencing.

PhD (D.C.) and MSc (A.P.D.O.T.) fellowships were funded by Coordenação de Aperfeiçoamento de Pessoal de Nível Superior (CAPES) Brazil, finance code 001.

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
