## [Reviewer comments · Microbiology Spectrum]

**Microbiology
Spectrum**

Emergence of thymidine-dependent *Staphylococcus aureus* small-colony variants in cystic fibrosis patients in Southern Brazil

Dilair Souza, Dilair Camargo, Laura Cogo, Libera Dalla-Costa, Ana Tomaz, DANIELI CONTE, Carlos Riedi, Nelson Rosário Filho, and Jussara Palmeiro

Corresponding Author(s): Jussara Palmeiro, Universidade Federal de Santa Catarina

Review Timeline:

Submission Date:

June 25, 2021

Accepted:

July 4, 2021

Editor: Joanna Goldberg

Reviewer(s): The reviewers have opted to remain anonymous.

Transaction Report:

DOI: <https://doi.org/10.1128/Spectrum.00614-21>

July 4, 2021

Dr. Jussara Kasuko Palmeiro
Universidade Federal de Santa Catarina
Departamento de Análises Clínicas
Florianópolis, SC
Brazil

Re: Spectrum00614-21 (Emergence of thymidine-dependent *Staphylococcus aureus* small-colony variants in cystic fibrosis patients in Southern Brazil)

Dear Dr. Jussara Kasuko Palmeiro:

With their revisions, the authors have adequately addressed the previous concerns of the reviewers.

Your manuscript has been accepted, and I am forwarding it to the ASM Journals Department for publication. You will be notified when your proofs are ready to be viewed.

Sincerely,

Joanna Goldberg
Editor, Microbiology Spectrum
